# Characterization among and within Sicilian Tetraploid Wheat Landraces by Gluten Protein Analysis for Traceability Purposes

**DOI:** 10.3390/plants13050741

**Published:** 2024-03-06

**Authors:** Samuela Palombieri, Marco Bonarrigo, Silvia Potestio, Francesco Sestili, Bernardo Messina, Giuseppe Russo, Claudia Miceli, Benedetto Frangipane, Marco Genduso, Chiara Delogu, Lorella Andreani, Stefania Masci

**Affiliations:** 1Department of Agriculture and Forest Science (DAFNE), University of Tuscia, Via San Camillo de Lellis Snc, 01100 Viterbo, VT, Italy; palombieri@unitus.it (S.P.); marco.bonarrigo@unitus.it (M.B.); silvia.potestio@unipr.it (S.P.); francescosestili@unitus.it (F.S.); 2Consorzio di Ricerca Gian Pietro Ballatore, Z.I. Dittaino, 90040 Assoro, EN, Italy; dinomessina@ilgranoduro.it (B.M.); giusepperusso@ilgranoduro.it (G.R.); 3Council for Agriculture Research and Economics, Plant Protection and Certification Center (CREA-DC), Palermo Headquarters, Viale Regione Siciliana Sud Est 8669, 90121 Palermo, PA, Italy; claudia.miceli@crea.gov.it (C.M.); benedetto.frangipane@crea.gov.it (B.F.); marco.genduso@crea.gov.it (M.G.); 4Council for Agriculture Research and Economics, Plant Protection and Certification Center (CREA-DC), Tavazzano Headquarters, SS9, Km 307, 26838 Tavazzano con Villavesco, LO, Italy; chiara.delogu@crea.gov.it (C.D.); lorella.andreani@crea.gov.it (L.A.)

**Keywords:** durum wheat, landraces, traceability, gliadin, glutenin subunits

## Abstract

The criteria of “Distinctness, Uniformity and Stability” as well as a high “overall quality index” are used to register the Italian modern varieties to the national register. Differently, local conservation varieties can be certified under different EU Directives that facilitate, as an overall objective, the preservation of biodiversity and the containment of genetic erosion. In recent years, products derived from ancient grains are perceived to be healthier and more sustainable by consumers, especially in Italy, with consequent higher market prices. The ancient tetraploid wheat varieties registered in the national register of conservation varieties amount to 28, 24 of which are Sicilian. They are supposed to have wide genetic variability compared to modern ones, making them vulnerable to fraud because they are difficult to trace. It is therefore important to have tools able to discriminate between autochthonous Sicilian varieties. This can be completed by gluten proteins composition, which also provides information on the technological properties of derived products. Fifty-one accessions belonging to twenty-two ancient varieties of Sicilian tetraploid (mostly durum) wheat were analyzed. Although wide intra-accession and intra-varietal variability measurements were assessed, the gliadin pattern of bulks of seeds belonging to each variety was discriminatory. Moreover, differences in technological attitudes were found between landraces. This paves the way to use gluten protein patterns for traceability, allowing local farmers and producers to valorize their products and assure consumers regarding the transparency of the entire supply chain.

## 1. Introduction

Wheat is one of the most cultivated crops in the world and provides a major source of nutrition globally. About 95% of the wheat grown worldwide is hexaploid bread wheat (*Triticum aestivum* L., 2n = 6x = 42, genome AABBDD), whilst the remaining 5% is mainly represented by durum wheat (*Triticum turgidum* ssp. *durum*, 2n = 4x = 28, genome AABB) [1]. The latter is well-adapted to semiarid environments and is mostly grown in the Mediterranean regions and North America.

After Canada, the second largest producer of durum wheat in the world is Italy, with 3.9 million tons (average data of the period 2019–2023), and more than half of the production comes from the southern regions [2,3]. The primary role of Italy in the cultivation of durum wheat is partly attributable to the use of semolina to produce traditional foods of the Mediterranean diet. In particular, pasta is a highly popular durum wheat-based food product because of its convenience, versatility, sensory, and nutritional values [4]. According to Italian law (Art. 6—DPR 187/2001), it can only be made with semolina. The economic importance of the pasta industry has fueled the intense breeding work carried out in Italy since the beginning of the 20th century. The main durum wheat breeding programs were focused on the release of elite varieties with higher grain yield and improved protein composition, strictly related to dough and pasta quality traits [5,6,7].

The spread of modern varieties caused a drastic reduction in the cultivation of landraces and old cultivars, which are characterized by lower yield and technological performance and general late maturity coupled with higher stature. For these reasons, they were grown for years in marginal areas and their conservation was delegated to public research institutes and custodian farmers. However, in recent years, there has been an increasing interest in old and landrace wheats, collectively known as “ancient wheats”, especially in Italy, because the consumers perceive their products to be more healthy, “natural”, and sustainable compared to modern ones. To date, the literature has shown conflicting results regarding the superior health properties of old varieties and landraces. Several studies demonstrated that some of these accessions have a higher content of phenols [8,9,10], a higher concentration of minerals [11], and increased prebiotic carbohydrates [12] compared to elite varieties. Other studies supposed that their superior beneficial effects were not due to their characteristic of being “old” but that they are more related to genotype and growth conditions [13,14,15,16,17]. To prove the beneficial-promoting attributes of ancient varieties, further investigations are required, with standard analysis methods on multiple genotypes of old and modern wheat grown in replicated multi-site field trials [18].

Nevertheless, the old cultivars and landraces represent a precious resource of genetic variability for traits associated with tolerance to (a)biotic stress and adaptation to different pedoclimatic conditions and low-input farming systems [19,20,21,22,23]. These favorable alleles or QTLs can be introgressed in elite varieties.

The landraces and historical varieties are populations that do not fall within the regulatory standards that define modern varieties today. In Italy, since 2009, these varieties “which are naturally adapted to the local and regional conditions and threatened by genetic erosion” can be registered in the National Catalogue of agricultural species (Legislative Decree No. 149 of 2009, now repealed and replaced by Legislative Decree No. 20 of 2021). This decree has the general objective of safeguarding biodiversity and limiting genetic erosion in agriculture. Nowadays, 28 varieties of tetraploid wheat are registered in the national catalogue as conservation varieties, and 24 of these are Sicilian.

In Sicily, the agriculture sector is of fundamental importance for the island economy and for the development of rural communities. Tetraploid wheat, including both *Triticum turgidum* ssp. *durum*, ssp. *turgidum* and ssp. *turanicum*, the latter also known as Khorasan wheat, represents one of the most important crops in terms of cultivated area (272,405 ha) [2], and some landraces are still grown in the mountain areas, constituting an important resource for low-input farming in marginal areas [24]. Although the old varieties are still produced, conserved, and exchanged among local farmers, the preservation of grain purity is often difficult during harvesting and post-harvesting procedures. Indeed, the lack of traceability along the supply chain (from seed to end product) does not guarantee the production and marketing of certified grain, increasing the risk of fraud and putting farmers’ economic profits at risk.

In this context, many efforts have been undertaken to preserve old varieties (i.e., Sicilian landraces) in purity and their traditional end products, with the aim of avoiding commercial fraud. Different strategies have been used in the development of traceability methods. Fiore et al. [25] used Cluster Analysis and Principal Component Analysis (PCA) obtained by single-nucleotide polymorphism (SNP) markers, together with agro-morphological, phenological, and quality-related traits, to classify and evaluate Sicilian ancient wheat germplasm. The same authors demonstrated that the SNP panel represents an efficient tool for the genetic traceability of old wheat varieties and can help to elude commercial fraud, sustaining the economic profits of the farmers. Fiore et al. [26] used the same strategy with a wider germplasm collection, including 75 accessions of Sicilian landraces, achieving the same results. Similarly, Taranto et al. [27] dissected the genetic variation patterns of two large germplasm collections of “Timilia” and “Russello” using SNP genotyping. Other molecular markers, such as RFLPs and SSRs, were successfully used for the characterization of durum wheat germplasm collections, enabling variations and resources of landraces to be more accessible for exploitation [28,29,30,31,32,33,34]. More analyses were carried out by markers based on morphological descriptors, storage protein composition, digestibility of starch, and concentration of secondary metabolites, permitting discrimination between new elite varieties and landraces [8,35,36,37,38,39,40].

Among biochemical markers, prolamins constitute an efficient system for the traceability of ancient wheat varieties due to their polymorphic nature [41,42]. Prolamins are storage proteins characterized by high proline and glutamine contents and distinguished in polymeric alcohol-insoluble (glutenins) and monomeric alcohol-soluble (gliadins) fractions. Both fractions constitute gluten, a visco-elastic network responsible for the rheological and technological quality of dough. Glutenins confer strength and elasticity to dough [43,44,45] and are formed by polymers stabilized by intra- and inter-chain disulfide bonds. The reduction of these bonds releases the individual glutenin subunits that have been classified in high-molecular-weight (HMW-GS) and low-molecular-weight (LMW-GS) glutenin subunits, on the basis of their molecular weight in Sodium Dodecyl Sulphate-PolyAcrylamide Gel Electrophoresis (SDS-PAGE) (reviewed by [46,47]). Gliadins confer extensibility and viscosity to dough and have been classified in three groups, known as α/β-, γ-, and ω-gliadins, based on their decreasing mobility in electrophoresis at acid pH [48,49]. In particular, durum wheat technological quality is mostly due to the presence of B-type low-molecular-weight glutenin subunits, with many subunits encoded by the *Glu-B3* locus. Two major groups of LMW-GS, designed LMW-1 and LMW-2, associated with γ-42 and γ-45 gliadins, were identified and associated with poor and good technological quality, respectively. In addition, both groups of subunits include components controlled not only at the *Glu-B3* locus but also at the *Glu-A3* and *Glu-B2* loci [50]. In both group variants, differences regarding the presence or absence of subunits have also been identified [51]. Although it is definitively clear that the technological quality is due to LMW-2 rather than to γ-gliadins, it is still not clear yet if the association between the presence of this allelic form and quality is due to structural or quantitative differences [52].

Visioli et al. [53] used HMW-GS as markers to analyze the genetic purity of grains and flours marketed and labelled as monovarietal. This study was carried out on four different tetraploid wheat Sicilian landraces (Timilia, Russello, and Margherito, which are durum wheats, and Perciasacchi, which is a Khorasan wheat) and highlighted numerous cross-contaminations by the other local varieties of durum and bread wheat grown on the same farm. They indicated that HMW-GS analysis was a useful marker to trace the varietal correspondence and genetic pureness of grains.

In this work, the biochemical profiles of prolamin proteins were used to trace the varietal correspondence and gluten quality of 51 accessions derived from 22 Sicilian landraces and old durum/tetraploid wheat varieties obtained by different custodian farmers in two different growing seasons.

## 2. Results

### 2.1. Analysis of Gluten Protein

The electrophoretic separations of the gluten components present in the Sicilian varieties were analyzed with the aim to evaluate the presence of heterogeneity in each accession. The A-PAGE and SDS-PAGE analyses of the 40 *T. durum*, 2 *T. turgidum*, and 9 *T. turanicum* accessions, listed in Appendix A, were carried out along with five varieties used as controls. To characterize the presence of intra-accession variability, ten spikes (two kernels for each) and a bulk sample were studied for each accession. Both electrophoretic analyses showed heterogeneity (Appendix A), thus indicating the presence of variability. The gliadin patterns showed greater differences than the glutenin ones and showed heterogeneity on 44 accessions of 51 studied (Appendix A). Instead, five accessions (Bivona, Ciciredda, Gioia, and two accessions of Perciasacchi) had a homogeneous profile for both gliadins and glutenins. The analysis of bulk samples confirmed the heterogeneity among accessions of the same variety both for gliadin and glutenin profiles (Appendix A).

To obtain a prolamin profile representative of each accession (here and after called “accession reference pattern”), all the kernels from ten spikes were pulled together. The electrophoretic analyses (Appendix A) detected inter-accession polymorphisms except for Scorsonera, Faricello, and Castiglione Glabro, which were uniform. Of the three accessions of Timilia Reste Bianche, one presented a different profile for gliadins (TRB-4), while Timilia Reste Nere showed inter-accession heterogeneity for both gliadins and LMW glutenin subunits (LMW-GS). Ruscia (synonymous with Russello Ibleo) showed two different profiles for gliadins and LMW-GS. Four accessions of Bidì were analyzed and one of them (BID-1) showed different profiles for gliadins, HMW-GS, and LMW-GS. The main differences were probably due to the mixture of two landraces differing at *GluB1* (20x + 20y and 13x + 16y), *GliA1*, and *GliA2* loci. Nine accessions of Perciasacchi (Khorasan wheat) showed two different composition profiles for both gliadins and glutenins. In particular, PER-3, PER-6, PER-8, and PER-13 had the typical composition of HMW-GS (Bx6 + By8), while PER-9 presented the subunit Bx20 + By20. The other accessions presented a mixed profile of the previous ones. Moreover, morphological analysis of these materials (data unpublished) revealed that two main types of Perciasacchi are grown by custodian farmers in Sicily, differing in terms of flowering time and beck morphology, although the most common seems to be the type showing Bx6 + By8 HMW-GS.

To obtain a representative profile of each landrace, a pool of kernels from all the studied accessions (here and after called landrace reference patterns) were analyzed (Figure 1). The electrophoresis analyses permitted to distinguish a unique storage protein composition for each landrace for both gliadins and glutenins. Timila Reste Bianche and Timilia Reste Nere showed the same gliadins pattern but different C-type LMW-GS. The only exceptions in which gliadins and glutenins failed to distinguish the unique profiles were for the landraces Bidì, Senatore Cappelli, and Capeiti 8, which showed similar profiles either for gliadins or glutenins.

In Table 1, the different gluten allelic compositions in the studied accessions are listed. Since LMW-GS attribution to a specific allelic form is susceptible to errors due to several protein bands with similar electrophoretic mobilities, we decided to attribute to the LMW-1 or LMW-2 types on the basis of the presence, in the latter, of the slowest moving protein band, named 42K LMW-GS, which characterizes this allelic form, whereas its absence identifies the LMW-1 types [52].

### 2.2. Measurement of the Polymeric Glutenin

UPPs are large unextractable polymers, which are related to dough properties. The recent breeding programs led to an improvement in the technological properties of wheat varieties, so it is expected to find a higher %UPP in all modern wheats.

SE-HPLC analysis was performed to quantify the amount and size of glutenin polymer in the grain of Sicilian tetraploid landraces Bidì, Castiglione Glabro, Gioia, Perciasacchi, Russello, Ruscia, Timilia Reste Bianche, Timilia Reste Nere, and two control durum wheat varieties, an old one, Senatore Cappelli, and a modern one, Saragolla.

Differences among wheat landraces/varieties and growing seasons were found for %UPP and Glu/Gli ratio (Table 2). In detail, as expected, the modern cultivar Saragolla had a higher %UPP. During 2020, all the genotypes analyzed showed an increase in the content of unextractable polymeric protein compared to the 2019 season except for the old cultivar, Senatore Cappelli, which showed a decrease of 21%, and the conservation varieties Bidì, Castiglione Glabro, Gioia, and Timilia Reste Bianche, in which the %UPP did not change. In both growing seasons, Ruscia showed the lowest content of %UPPs, whereas Russello showed the highest among Sicilian landraces. The %UPP value of the latter one was close to modern variety Saragolla. Instead, Perciasacchi had the largest difference in %UPP (>65%) between the growing seasons of 2019 and 2020.

The Glu/Gli ratio was influenced by the growing season differently than the %UPPs (Figure 2). The conservation varieties Bidì and Gioia showed the lowest ratio of Glu/Gli during 2019. Conversely, during the 2020 season, Bidì showed the highest value, even more than the modern variety Saragolla. Castiglione Glabro, Timilia Reste Bianche, and Timilia Reste Nere demonstrated worsening in the Glu/Gli ratio in the 2020 season compared to the previous one. Finally, Saragolla confirmed good technological quality during the season 2020, while Senatore Cappelli was the worst.

The percentages of sums of squares accounting for the effects of the growing season were 12% and 2% for %UPP and Glu/Gli, respectively. The greatest contribution to the difference in Glu/Gli ratio was provided by the interaction of variety x year. Instead, the varietal effect contributed to 52% and 12% for %UPP and Glu/Gli ratio, respectively (Figure 2).

## 3. Discussion

There has been increasing interest in wheat landraces in recent years, especially in Italy, where tetraploid wheats, particularly durum wheat, have a long tradition of growing and breeding. Sicily is the region that contributes most, with 24 out of the 28 landraces recorded in the National Register of Conservation Varieties. This germplasm is agronomically and nutritionally interesting and represents a valuable resource to preserve cereal genetic diversity. Products based on wheat landraces, so-called “ancient Sicilian grains”, are present on the shelves of large Italian retailers. However, correspondence with variety is not guaranteed due to a lack of traceability along the supply chain. It is thus important that such landraces are deeply and clearly characterized and traceable.

Within this context, our work aimed to characterize 51 accessions belonging to 22 Sicilian tetraploid wheat landraces and to assess the intra- and inter-accession variability. Since each cultivar has a specific prolamin composition, electrophoretic patterns of both glutenins and gliadins were used as characterization criteria. Several studies have demonstrated that the electrophoretic profile of prolamins is specific for each cultivar and is not influenced by growing conditions, kernel protein content, or incomplete seed maturation [54,55,56].

Analyses of individual spikes allowed to detect heterogeneity in the same accession, mostly revealed by gliadins pattern. Indeed, 85% of the studied accessions differed by allelic composition in at least one *Gli* locus. This was expected as gliadins are one of the most polymorphic proteins in nature, with 147 alleles in tetraploid wheat identified to date [54]. Cases of intra-varietal non-uniformity of common wheat cultivars have been previously described [56,57,58,59]. This variability could be explained as the landraces were obtained by crossing different wheat genotypes that were characterized by different allelic compositions at the *Gli* loci [60] and were not selected for this trait. For this reason, the different ecotypes that constitute a landrace may contribute to its adaptivity, and thereby to the known high plasticity of the local varieties.

The accessions’ reference patterns showed different compositions of HMW-GS in the Gioia, Russello, and Perciasacchi landraces. For Gioia, it seems to be a contamination because the different bands were faint and present only in one accession (Appendix A). The Perciasacchi accessions showed the presence of Bx20 + By20 or Bx6 + By8 subunits. The first HMW-GS composition was already described by Visioli et al. [53] in the pure seeds provided by the CREA’s seed bank. In this study, the authors analyzed commercial flour labelled as monovarietal and found contaminations regarding the HMW-GS subunits. Most of the samples are characterized by a different profile, with the presence of subunits Bx6 + By8. Our data confirmed that the form showing Bx6 + By8 HMW-GS is the most common in Sicilian Perciasacchi landraces grown by custodian farmers and showed how the institutional seed banks often do not reflect the genetic variation of the actually grown materials.

In the same study [53], the analysis of the flours obtained by Russello showed the HMW-GS composition involving Bx13 + By16 and Bx6 + By8. The presence of Bx6 + By8 was explained by the authors as a contamination with Ruscia. In this paper, the accessions of the same landrace (Russello) showed polymorphism with the presence at the loci *Glu-A1* (null or 2*) and *Glu-B1* (Bx6 + By8 or Bx13 + By16 or both) gluten subunits. In our study, all the accessions of Ruscia highlighted a different composition at *Glu-B1* (Bx20 + By20) compared to those described by Visioli et al. [53] (Bx6 + By8), while the same composition (Bx20 + By20) was observed for Bidì (Margherito). In agreement with our results, another work confirmed the presence of the Bx13 + By16 subunits for Russello, while the same authors described a different HMW-GS (Bx20 + By20 vs. Bx6 + By8 subunits) for Timilia Reste Bianche [35].

The landrace reference patterns showed a different composition in prolamins profiles for all the ancient wheats, confirming that electrophoretic analysis of gliadins and glutenins is a valid method for the traceability of these materials. The only exceptions were for Bidì, Senatore Cappelli, and Capeiti 8, which showed the same prolamin composition. The similarity of gliadins and glutenins profiles is explained by the common origin shared by these varieties. In particular, Bidì and Senatore Cappelli are very similar morphologically, and they were reported to be selected, respectively, by Tunisian landraces Mahmoudi and Jean Retifah; Capeiti 8 was obtained by the cross of Eiti 6 x Senatore Cappelli [5,34,61]. However, De Cillis [62] reported that both Bidì and Senatore Cappelli were selected from the same Tunisian landrace named Bidì, and, recently, some other studies supported this hypothesis [26,27,63]. To trace these landraces requires a different method, such as SNP genotyping, which was useful for revealing genetic diversity in a large population of two Sicilian landraces, Russello and Timilia [27].

Moreover, in this paper, analysis of storage proteins enabled comparing dough quality among landraces in two different growing seasons. Gluten polymers contribute to the technological quality of the wheat by influencing the rheological properties of the dough, and this can be predicted by the amount of %UPP and Glu/Gli [50,64]. The amount of insoluble polymeric proteins (%UPP) was higher for all the accessions analyzed during the 2020 season due to favorable growing conditions except for Senatore Cappelli, which showed a decrease in these proteins. Furthermore, the low %UPP found in this variety was predictable due to the presence of Bx20 + By20 HMW-GS, correlated with low technological quality [65]. Instead, during the 2019 growing season, Bidi showed a similar % UPP to the modern variety Saragolla and, during the 2020 growing season, a higher Glu/Gli ratio despite having the same HMW composition of Senatore Cappelli. Differences in %UPP between landraces with the same allelic composition were due to differences in storage protein subunit amount and could be explained by differences in gluten strength [35]. Therefore, the 2019 season was characterized by adverse growing conditions, and the Russello variety evidenced possessing a technological quality close to the modern variety Saragolla, although, when the environmental conditions are favorable (as in 2020), the Saragolla values are higher, as expected. Instead, Perciasacchi is the genotype that suffered the most from the adverse weather conditions and had the largest difference in %UPP between the two growing seasons.

## 4. Materials and Methods

### 4.1. Plant Material and Experimental Conditions

The collection of Sicilian grains included in this study are 19 *Triticum turgidum* subsp. *durum* (durum wheat), two *Triticum turgidum* subsp. *turgidum*, and one *Triticum turgidum* subsp. *Turanicum* (khorasan wheat) for a total of 51 accessions (Appendix A); two historical cultivars (Senatore Cappelli and Trinakria) and three modern durum wheat cultivars (Iride, Core, and Saragolla) were used as controls. The cultivation of these materials was carried out in Sicily in two different locations in 2019 (Palermo, 38°06′54.36″ N; 13°21′02.88″ E. 46 m a.s.l.) and in 2020 (Vicari, Palermo province, 37°49′29″28 N., 13°34′1″20 E., 640 m a.s.l.) under conventional agricultural regime.

Initially, in gliadin and glutenin gels, the extract from two seeds from each of ten spikes, along with a bulk of seeds obtained from each accession, were run. Subsequently, to compare accessions with each other, gels were prepared using the bulk samples of all accessions from all varieties. Finally, the bulk samples of all accessions from each variety were mixed to obtain a representative bulk of the varieties, which was then used to produce the final gels.

### 4.2. Extraction of Gluten Protein

Two seeds from each sample spike and five seeds for each sample bulk of the different wheat landraces and commercial varieties were crushed in a mortar with a pestle until a fine powder was obtained. The procedure for extracting gluten protein followed the sequential extraction protocol. The gliadins were extracted from 15 mg of whole flour in 1.5 M dimethyl formamide (1:10, *w*/*v*) with continuous mixing for 1 h at room temperature, followed by centrifugation at 14,000× *g* for 10 min. The pellet was stored at −20 °C and subsequently used for the extraction of the glutenin subunits while the supernatant containing the gliadins fraction was collected. The gliadin loading dye (50% saccharose (*w*/*v*) and 0.02% (*w*/*v*) dimethyl violet) was added to each sample. The tubes were then vortex-mixed, centrifuged for 2 min at 14,000× *g*, and 10 μL of the supernatant were used to load the gels.

Extraction of glutenin fraction was conducted as described by Ibba et al. [66] with minor modifications. Briefly, the pellet obtained from the first dimethyl formamide treatment was washed two times with 750 μL of 50% 1-propanol and, after centrifugation, the supernatant was discarded to remove any remaining gliadins from the pellet. A volume of 100 μL of a solution with dithiothreitol (DTT) at 1.5% (*w*/*v*) formed with 50 μL of 1-propanol at 50% (*v*/*v*) and 50 μL of Tris-HCl 0.08 M pH 8.0 was then added to the pellet. The tubes were mixed in a vortex and incubated for 30 min at 65 °C for the extraction of glutenin fraction. After the centrifugation at 10,000× *g* for 2 min, 100 μL of a solution with vinylpyridine at 1.4% (*v*/*v*) formed with 50 μL of propanol at 50% (*v*/*v*) and 50 μL Tris-HCl 0.08 M pH 8.0 was added to the tubes, which were then mixed with a vortex, incubated for 15 min at 65 °C, and centrifuged for 2 min at 14,000× *g* for alkylation of glutenin subunits. The supernatant containing the glutenin fraction was recovered and transferred to new tubes containing the loading solution 2× (Tris-HCl pH 6.8, 2% SDS, 40% glycerol and 0.02% bromophenol). The samples were mixed in a vortex, incubated for 5 min at 90 °C, and then centrifuged for 2 min at 14,000× *g*, and 10 μL of the supernatant were loaded on polyacrylamide gels.

### 4.3. Electrophoretic Separation

A-PAGE was performed using polyacrylamide gels (8 × 13 cm) with 8% of T value (total acrylamide concentration) and 1.25% of C value (cross-linker concentration) containing 40 mM aluminum lactate, 260 mM lactic acid, and 0.002 g of ascorbic acid. After electrophoretic separation at 25 mA, the gels were stained with a Commassie R-250 solution fixed in 5% ethanol and 12% trichloroacetic and de-stained in deionized water.

SDS-PAGE was performed using separating gel with 15% T value and 1.3% of C value. The main gel was prepared using 0.36 M Tris-HCl, pH 8.8, and 0.1% SDS. Gels were run at 12.5 mA per gel for 20 h and stained with brilliant blue G-colloidal solution fixed in 20% methanol and 10% phosphoric acid and de-stained in deionized water.

The two biotypes of the Italian durum wheat cultivar Lira 42 and Lira 45 were used as standards of γ-gliadins γ-42 and γ-45, as well as of LMW-1 and LMW-2 types. As HMW-GS standards, the Italian durum wheat cultivars Core (7 + 8), Iride (7 + 8), Saragolla (6 + 8), and Senatore Cappelli (20x + 20y) were used.

### 4.4. Analysis of Unextractable Polymeric Proteins (%UPP) by SE-HPLC

SE-HPLC was used to measure the percentage of non-extractable polymer proteins (%UPP). The extraction of the SDS-soluble fraction and the SDS-insoluble fraction was performed as reported by Gagliardi et al. [67], except for the sonication, which was performed with the probe-type sonicator SONICS Vibracell model VC 50T (power 50 W, frequency 20 KHz) for 15 s. Briefly, soluble fraction was first extracted from semolina using SDS-phosphate buffer. The remaining pellet was resuspended in the same buffer and sonicated to obtain the UPP portion of semolina protein. The HPLC equipment used was a HPLC Knauer Smartline system equipped with two Smartline 1000 pumps, one HT300L autosampler, one Smartline 2600 UV detector, and a TSKgel SuperSW3000 column (300 mm × 4.6 mm). The flow rate was 0.35 mL/min, and the detection was at 214 nm. Every sample was analyzed in two replicates. The area under the first peak of the soluble fraction was called P1_s_, while the one under the other peaks was named P2_s_. The areas of the insoluble fraction were named, respectively, P1_ns_ and P2_ns_ (Appendix A). The %UPP and the Glu/Gli ratio have been determined as
$$\%UPP=\left(\frac{{P1}_{ns}}{{P1}_{ns}+{P1}_{s}}\right)\times 100$$
$$\frac{Glu}{Gli}=\frac{{P1}_{ns}+{P1}_{s}+{P2}_{ns}}{{P1}_{s}}$$

### 4.5. Statistical Analysis

The %UPP and Glu/Gli values of tetraploid varieties cultivated in the same year were analyzed by one-way analysis of variance (ANOVA), and pairwise analysis was carried out using the Tukey Honestly Significant Difference test (Tukey test) (*p* < 0.05). The comparison of these values between the two years was analyzed with Student’s *t*-test (*p* < 0.05). For the calculation of the Sum of Squares, a two-way ANOVA was performed.

All statistical processing was completed with R, version 4.1.3 [68].

## 5. Conclusions

Ancient Sicilian wheat holds significant importance in producing traditional niche products that fuel the local economy. Despite a growing demand for tetraploid cereal products in the traditional retail market, limited production raises concerns about potential fraud. The current lack of traceability throughout the supply chain poses a challenge to ensure alignment between certified seeds and final products. Semolina labeled as monovarietal or its derived products may not accurately represent the declared variety due to this traceability gap. Additionally, the conservation of these wheat varieties relies on custodian farmers, and these populations often exhibit a level of heterogeneity.

Our study indicates that the biochemical profile of prolamins effectively distinguishes among ancient tetraploid wheat varieties, addressing traceability issues to the mutual benefit of consumers and producers. Furthermore, the comparison among accessions by custodian farmers revealed intra- and inter-accession heterogeneity for prolamin profiles. In fact, these landraces have been naturally selected for agronomic traits and not for gluten quality. Particularly, the polymorphic nature of gliadin enables more distinct discrimination among different accessions.

In conclusion, the biochemical profile of prolamins represents a valid tool to trace ancient wheats, contributing to the preservation of biodiversity and avoiding fraud along the supply chain.

## Figures and Tables

**Figure 1 plants-13-00741-f001:**
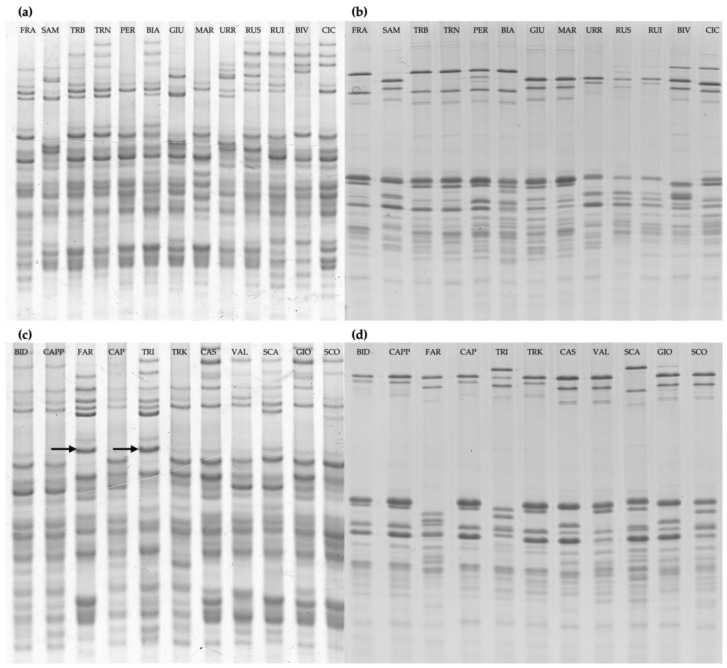
Electrophoretic separation of gluten proteins of the varietal bulks. (**a**,**c**) A-PAGE analysis of gliadins fractions. (**b**,**d**) SDS-PAGE analysis of glutenins fraction. FRA, Francesa; SAM, Sammartinara; TRB, Timilia reste bianche; TRN, Timilia reste nere; PER, Perciasacchi; BIA, Biancuccia; GIU, Giustalisa; MAR, Martinella; URR, Urria; RUS, Russello; RUI, Ruscia; BIV, Bivona; CIC, Ciciredda; BID, Bidì; CAPP, Senatore Cappelli; FAR, Faricello; CAP, Capeìti; TRI, Tripolino; TRK, Trinakria; CAS, Castiglione Glabro; VAL, Vallelunga; SCA, Scavuzza; GIO, Gioia; SCO, Scorsonera. The arrows indicate the γ-42 associated with the presence of LMW-1 in the varieties Faricello e Tripolino. All the other varieties show allelic forms of LMW-2 type (**a**,**d**).

**Figure 2 plants-13-00741-f002:**
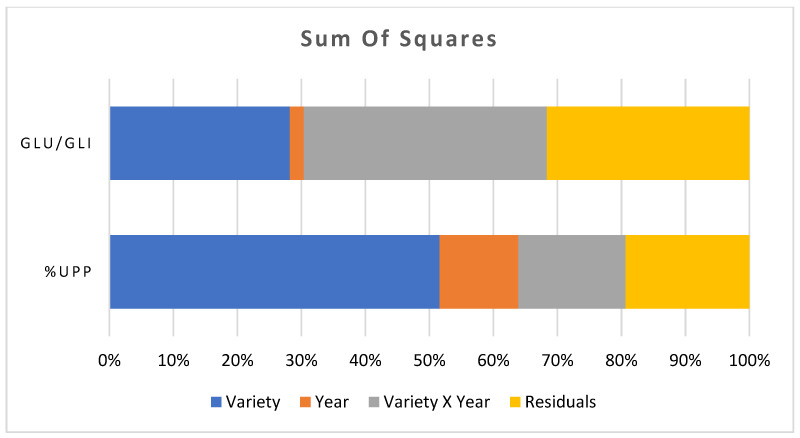
Percentage of the total sums of squares for the effects of variety, year, interaction between variety and year, and residuals.

**Table 1 plants-13-00741-t001:** Details of the allelic composition of gluten genes of Sicilian landraces. HMW-GS coded by *Glu-A1* and *Glu-B1* loci are listed in sequence.

ID_CREA	Conservation Varieties	HMW-GS	LMW-GS Type	γ-Gliadins
FD-BIA-1	Biancuccia	null, 6 + 8	2	42
FD-BID-1	Bidì	null, 20x + 20y	2	45
FD-BID-2	Bidì	null, 20x + 20y	2	45
FD-BID-3	Bidì	null, 20x + 20y	2	45
FD-BID-5	Bidì	null, 20x + 20y	2	45
FD-BIV-1	Bivona	2*, 13 + 16	1	44
FD-CAP-1	Capeiti 8	null, 20 + 20	2	45
FD-CAS-2	Castiglione Glabro	null, 13 + 16	1	45
FD-CAS-3	Castiglione Glabro	null, 13 + 16	1	45
FD-CIC-1	Ciciredda	2*, 32 + 33	2	45
FD-FAR-1	Faricello	null, 13 + 16	1	42
FD-FAR-2	Faricello	null, 13 + 16	1	42
FD-FRA-1	Francesa	null, 6 + 8	2	45
FD-GIO-1	Gioia	null, 6 + 8/13 + 16	1	45
FD-GIO-2	Gioia	null, 13 + 16	1	45
FD-GIU-1	Giustalisa	null, 13 + 16	2	47
FD-MAR-1	Martinella	null, 13 + 16	2	45
FD-PER-1	Perciasacchi	null, 6 + 8/20x + 20y	2	45
FD-PER-2	Perciasacchi	null, 6 + 8/20x + 20y	2	45
FD-PER-3	Perciasacchi	null, 6 + 8	2	45
FD-PER-4	Perciasacchi	null, 6 + 8/20x + 20y	2	45
FD-PER-6	Perciasacchi	null, 6 + 8	2	45
FD-PER-7	Perciasacchi	null, 6 + 8/20 + 20	2	45
FD-PER-8	Perciasacchi	null, 6 + 8	2	45
FD-PER-9	Perciasacchi	null, 20x + 20y	2	45
FD-PER-13	Perciasacchi	null, 6 + 8	2	45
FD-RUI-1	Ruscia	null, 20x + 20y	2	45
FD-RUI-3	Ruscia	null, 20x + 20y	2	45
FD-RUI-4	Ruscia	null, 20x + 20y	2	45
FD-RUS-1	Russello	2*, 13 + 16/6 + 8	1	45
FD-RUS-3	Russello	2*, 13 + 16/6 + 8	1	45
FD-RUS-4	Russello	null, 13 + 16/6 + 8	1	45
FD-RUS-6	Russello	null, 13 + 16	1	45
FD-SAM-1	Sammartinara	null, 13 + 16	2	47
FD SCA-1	Scavuzza	null, 6 + 8	2	44
FD-SCO-1	Scorsonera	null, 20	2	45
FD-SCO-2	Scorsonera	null, 20	2	45
FD-TRB-1	Timilia R.B.	null, 6 + 8	2	44
FD-TRB-2	Timilia R.B.	null, 6 + 8	2	44
FD-TRB-4	Timilia R.B.	null, 6 + 8	2	44
FD-TRI-1	Tripolino	null, 6 + 8/13 + 16	1	42
FD-TRN-1	Timilia R.N.	null, 6 + 8	2	44
FD-TRN-3	Timilia R.N.	null, 6 + 8	2	44
FD-TRN-6	Timilia R.N.	null, 6 + 8	2	44
FD-TRN-7	Timilia R.N.	null, 6 + 8	2	44
FD-TRN-8	Timilia R.N.	null, 6 + 8	2	44
FD-TRN-9	Timilia R.N.	null, 6 + 8	2	44
FD-TRN-11	Timilia R.N.	null, 6 + 8	2	44
FD-TRN-12	Timilia R.N.	null, 6 + 8	2	44
FD-URR-1	Urria	null, 20x + 20y	2	47
FD-VAL-2	Vallelunga pubescente	null, 13 + 16	2	45

**Table 2 plants-13-00741-t002:** Content of %UPP and Glu/Gli ratio of different landraces related to the two growing seasons.

		%UPP		Glu/Gli	
Name	Type	2019	2020	2019	2020
Bidì	Landrace	33.9 ± 1.0 ab	38.8 ± 2.6 bc	0.480 ± 0.021 c	0.734 ± 0.040 a *
Castiglione Glabro	Landrace	29 ± 1.2 abc	28.1 ± 0.3 de	0.639 ± 0.042 ab *	0.412 ± 0.026 d
Gioia	Landrace	29.4 ± 1.1 abc	31.2 ± 0.8 cde	0.465 ± 0.037 c	0.556 ± 0.004 bcd
Perciasacchi	Landrace	25.9 ± 1.7 c	38.9 ± 3.0 bc *	0.586 ± 0.024 bc	0.505 ± 0.031 cd
Russello	Landrace	36.9 ± 1.2 a	44.2 ± 2.0 ab *	0.622 ± 0.018 ab	0.573 ± 0.029 bc
Ruscia	Landrace	15.1 ± 0.5 d	25.2 ± 1.3 de *	0.560 ± 0.016 bc	0.492 ± 0.056 cd
Saragolla	Modern Cultivar	30.8 ± 3.4 abc	50.7 ± 2.2 a *	0.580 ± 0.031 bc	0.664 ± 0.017 ab *
Senatore Cappelli	Old Cultivar	28.3 ± 1.6 bc *	22.4 ± 1.9 e	0.723 ± 0.017 a *	0.559 ± 0.014 bcd
Timilia Reste Bianche	Landrace	30.1 ± 0.7 abc	30.3 ± 1.0 cde	0.530 ± 0.022 bc *	0.424 ± 0.021 d
Timilia Reste Nere	Landrace	27.2 ± 1.4 bc	33.5 ± 1.1 cd *	0.519 ± 0.031 bc *	0.428 ± 0.018 d

For each landrace, means within columns followed by different lower-case letters refer to the statistical analysis performed using one-way analysis of variance (ANOVA) with post-hoc Tukey HSD test (*p* < 0.05). For each sample, means within raw followed by asterisks indicate a significant difference in the two different years. The statistical analysis was performed using Student’s *t*-test (*p* < 0.05).

## Data Availability

Data are contained within the article and Appendix A.

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
