# Peer review of "Characterization among and within Sicilian Tetraploid Wheat Landraces by Gluten Protein Analysis for Traceability Purposes"

_plants, 2024, doi:10.3390/plants13050741_

Round 1
Reviewer 1 Report
Comments and Suggestions for Authors After reading the manuscript , I realized that the manuscript showed in some parts the scientific rigour wanted, but in other parts I have missed it.The authors have presented critical evaluation only in some paragraphs.
The references are not exactly current, besides the objective should be adjusted.
Thats why I have written some comments below in an attempt to improve the paper.
Suggestion: Following your objectives, the title could be something on this way: Biochemical profile of prolamins protein and gluten quality of Sicilian Tetraploid Wheat Landraces
L.44- Sensory instead of sensorial
L.45- Why? Only semolina, please add this information.
L.58- Since you mentioned " healthy properties" - any comments about celiac disease ?
L.63- Again " healthy properties "
L139- 141- "accessions derived from 22 Sicilian landraces and old durum/tetraploid wheat varieties obtained by different custodian farmers in two different growing seasons." =>Some information in this sentence are Material and methods, not necessary here.
L.142 - Results ??
L.198- It seems to me that table 1 ( or in other table) could also contain the physical characteristics, at least color, size ... main peculiarities.
L.226- This result wasn't covered much.
L.229- Since the two growing seasons are important, shouldn't they be have evaluated statitically as well ?
L.318- The section Material and methods is out of the place.
L.325- Wouldn't it be important to include the harvest month as well?
L.325- "in two different locations" - characteristics of both soils ?
L.327- I think it's essential to include the temperature average for both years, as well as the rainfall rate.
L.328- It seems to me that we missed the moment when the harvest occurred and when the samples were taken to the laboratory for analysis.
Ex: same day ? samples frozen ?
repetitions ?
Which design? Completely randomized? Randomized blocks ? ....
L.334- Extraction of Gluten Protein - could be in the title/ objective.
L.409- The conclusion could be expanded. You have repeated the final part of the objectives.
L. 362- " Electrophoretic Separation " - author/ year
L.379- Unfortunately, material and methods was inserted in the wrong place in the paper and all the authors will need to be re-numbered.
Comments on the Quality of English LanguageMinor editing of English language required
Author Response
The answers (in red) are reported in the uploaded file

Reviewer 2 Report
Comments and Suggestions for Authors
Review comments on
Characterization Among and Within Sicilian Tetraploid Wheat Landraces by Grain Storage Protein Analysis
Developing a method for tracking wheat genes and composition, to improve wheat yield, kernel quality etc. has been the important research subject since centuries. This includes searching for the indices that will allow to find the relations between wheat characteristics and their breeding and processing outcomes, or to build market dominance. Shortly, the idea is not original nor new. However, Authors made a wide research on majority of old wheat varieties planted in Sicilia. It constitutes a valuable material to discriminate between durum wheats to preserve old Sicilian landraces.
The objective of the study as indicated by authors - “It is therefore important to have tools able to discriminate the autochthonous Sicilian varieties” and “our work aimed to characterized 51 accessions belonging to 22 Sicilian tetraploid wheat landraces and to assess the intra- and inter-accession variability - was clearly stated and followed by the applied procedures and analyses done. However, the conclusions do not correspond to the objective of the work. They don’t results from the research. The first part sound like the introduction, to the problem, the second is very general and it is rather obvious that it is possible to use %UPP to predict technological quality”. I expect that the conclusions will relevantly adhere to the objective (find the tool), observed from results potential possibilities, advantages, and disadvantages, or future study needs.
The paper is well written and carefully edited. There are only few editorial mistakes, like missing spaces (line 126, 127, 143 (dot), 159 (e?), 186, 262, 267, 269, 328.
In summary. The manuscript can be published in Plants with very minor changes.
Author Response
Thanks for the observation and suggestion.
The conclusion was re-written and the editorial mistakes have been corrected.
Round 2
Reviewer 1 Report
Comments and Suggestions for Authors After another evaluation of the manuscript, I realizes great improvement in the quality of the paper. The authors have accepted some of my requests, but they have justified with confidence.They also have improved the English, however it is always useful to ask a native speaker for a final appreciation.
Comments on the Quality of English LanguageMinor editing of English language required